# Shaping the future of medical education: A cross-sectional study on ChatGPT attitude and usage among medical students in Sudan

**Weam Mohamed Meargni Ahmed***, **Malaz M. Abdalmotalib** *, **Mohamed H. Elbadawi, Galia Tajelsir Fadulelmula Mohammed, Waad Mohamed Ibrahim Mohamed, Fatima Salih Babiker Mohammed, Hajar Saad Salih, Hiba Omer Yousif Mohamed**

Faculty of Medicine, University of Khartoum, Khartoum, Sudan.

* malaz.abdalmotalib@gmail.com (MMA); weaammohamed376@gmail.com (WMMA)

## Abstract

### Background

Artificial intelligence (AI) is revolutionizing education globally, yet its adoption in medical education remains inadequately understood. ChatGPT, a generative AI tool, offers promising yet doubtful potential for enhancing academic and clinical training.

### Methods

This study employed an analytical cross-sectional design, involving 1,443 Sudanese medical students who participated through an online, structured questionnaire. The questionnaire was designed to assess ChatGPT awareness, usage, and associated factors. Statistical analysis was performed using SPSS software to identify key determinants influencing ChatGPT awareness and usage among the participants.

### Objective

This study investigates the levels of awareness, attitude, and usage of ChatGPT among Sudanese medical students, identifying key socio-demographic, economic, and institutional factors influencing its adoption.

### Results

Among the participants, 65.8% were aware of ChatGPT, yet only 41.9% reported using it. Gender differences were statistically significant, with males demonstrating higher usage rates ($p < 0.001$). Single and unemployed students were more likely to use ChatGPT compared to their counterparts. Students residing in private accommodations and those with higher family incomes (>300,000 SDGs) showed significantly greater usage ($p < 0.001$). Factors such as residency type, internet quality, and institutional orientation were identified as key influences on ChatGPT adoption, highlighting

**Data availability statement:** All relevant data are within the manuscript and its Supporting Information files.

**Funding:** The author(s) received no specific funding for this work.;

**Competing interests:** The authors have declared that no competing interests exist.

**Abbreviations:** AI, Artificial intelligence; Chat(GPT), Generative Pre-trained Transformer; IGCSE, The International General Certificate of Secondary Education; SDG, Sudanese Pound.

a substantial digital influence. Additionally, inadequate infrastructure and reliance on traditional curricula were significant barriers to wider usage.

## Conclusions

The findings underscore the urgent need for targeted interventions, including curriculum reform to integrate AI literacy, enhanced digital infrastructure, and gender-equity initiatives. Addressing these systemic gaps will scale up AI adoption in medical education. This study provides actionable insights for educators and policymakers, emphasizing the urgency of bridging socio-economic and institutional inequities to foster equitable access to AI tools in medical training.

## Introduction

Artificial Intelligence (AI) refers to a broad array of technologies designed to simulate human intelligence through machines, such as computers and robots [1]. Over the past decade, AI has witnessed rapid advancements, becoming increasingly pervasive across multiple sectors, including healthcare, education, and various other industries [2]. Its applications in medical education have sparked considerable interest, driven by the growing need to enhance learning experiences, clinical decision-making, and patient care [3]. Among the most notable breakthroughs in AI is the advent of conversational AI models like ChatGPT, which have significantly reshaped how both students and professionals interact with information. These tools leverage natural language processing (NLP) to generate human-like, coherent responses to user inputs, thereby enhancing accessibility to knowledge and fostering more interactive and personalized learning environments [4].

Built on OpenAI's GPT-3.5 architecture, which comprises over 175 billion parameters, ChatGPT demonstrates significant potential in the field of medical education [2,5,6]. It offers rapid access to medical knowledge, aids in clinical decision-making, and delivers personalized educational content to address knowledge deficiencies. Additionally, its ability to analyze large volumes of scientific literature and produce succinct summaries enhances its value as a resource for medical research [7,8].

Despite its promise, ChatGPT presents several challenges. Accuracy remains a concern, as AI-generated content can sometimes be misleading, especially in high-stakes clinical contexts [9–15]. Ethical concerns also arise, including academic dishonesty, misinformation, and over-reliance on AI tools [16]. Moreover, effective implementation is often constrained by disparities in access to digital infrastructure and the broader systemic issue of the digital divide.

In developing countries like Sudan, the integration of AI into education faces unique barriers. Although AI holds potential to address issues like diagnostic errors, workload, and resource inefficiencies [17], numerous obstacles hinder its adoption. These include limited financial and technical resources, lack of skilled professionals, legal and data privacy concerns, and social skepticism [17–19]. Studies on e-learning in Sudan have also documented the impact of restricted internet access, political

instability, and limited institutional support [20,21], all of which directly affect student engagement with digital tools like ChatGPT.

While prior research has explored Sudanese medical students' general awareness and attitudes toward AI [22], there is a notable lack of studies examining the use of ChatGPT specifically within medical education. Given the tool's familiarity and adoption globally, there is a pressing need to understand its application and impact in under-resourced contexts such as Sudan [23].

ChatGPT has a privilege over specialized AI tools like Med-PaLM and Glass Health due to its widespread availability and usage across both academic and non-academic settings. Qualitative studies in similar educational environments, such as in Saudi Arabia, have shown high levels of awareness and adoption of ChatGPT among students and faculty [23]. Research further demonstrates its integration into medical education for tasks like literature summarization and research writing [24–27].

This study aims to explore the awareness, usage, and attitudes toward ChatGPT among Sudanese medical students. Special attention is given to how factors such as gender, socio-economic background, academic year, and technological access influence their engagement with this tool. Understanding these dynamics is critical for promoting equitable AI integration, contributing to the broader discourse on responsible and inclusive AI implementation, and ensuring that all students, regardless of their circumstances, can benefit from technological advancements in education.

## Methods and materials

### Study design

This study employed an analytical cross-sectional design to evaluate the awareness, attitude, and usage of ChatGPT among medical students in Sudan. Data were collected from June 20th, 2024, to October 30th, 2024, ensuring adequate time to capture a diverse and representative sample.

### Study area and population

This study encompassed medical students from a wide spectrum of public and private universities across Sudan, ensuring a diverse and representative sample reflective of varying educational contexts. **Public universities** included prominent institutions such as the University of Khartoum, Gezira University, Nile Valley University, Kassala University, Shendi University, Al-Zaeem Al-Azhari University, Omdurman Islamic University, and Sudan University of Science and Technology. On the **private sector** side, institutions like Al-Razi University, Al-Qalam College for Science and Technology, Al-Mashriq University, the University of Science and Technology, Al-Fajr University, Al-Bayan University, and Al-Ahfad University for Women contributed to the study. The study employed specific eligibility criteria. Medical students from all academic years, aged 18 years or older, were included. Students younger than 18 years, graduates, and those pursuing non-medical degrees, regardless of any prior medical background, were excluded

### Sample size and sampling technique

The sample size was calculated using Cochran's formula for the unknown population [28]

$$n = \frac{z^2 \cdot p \cdot (1-p)}{e^2}$$

adopting a 95% confidence level (Z = 1.96), an estimated population proportion (p) of 0.5 to account for maximum variability, and a margin of error (e) of 0.05. The minimum required sample size (n) was calculated to be 385 participants. However, the study recruited 1,443 respondents to compensate for the limitations associated with the use of a convenience sampling technique.

## Data collection tool

Data were collected using a validated questionnaire, which was originally developed by Sallam et al. (2023) to assess ChatGPT awareness, attitudes, and usage among health students in Jordan [29]. We used the same questionnaire without any further modifications. The questionnaire was divided into three main sections:

1. **Declaration and consent:** The survey began with a comprehensive explanation of its objectives, accompanied by a mandatory electronic consent requirement for successful participation. The introduction emphasized participant anonymity and data privacy by explicitly stating that no personal identifiers, such as names or email addresses, would be collected

2. **Socio-demographic information:** Participants provided data on age, gender, marital status, residency, employment, family income, university, type of admission, academic year, academic performance, and extracurricular activities, allowing diverse and estimated insights of chat GPT adoption.

3. **ChatGPT attitude and usage:** Participants were then asked whether they had heard of ChatGPT prior to the study. A "yes" response allowed them to proceed to the next question, while a "no" response led to survey submission. Subsequently, participants were asked whether they had used ChatGPT prior to the study. Those who responded "yes" were presented with the full set of 36 items, whereas those who answered "no" received only the first 13 items based on the Technology Acceptance Model (TAM) that was better clarified by a study conducted by Sallam et al. (2023) [29]. All items were assessed on a 5-point Likert scale, where "strongly agree" was scored as 5, "agree" as 4, "neutral/no opinion" as 3, "disagree" as 2, and "strongly disagree" as 1. For items reflecting a negative attitude toward ChatGPT, reverse scoring was applied to negatively worded items. Attitude about ChatGPT was recorded using 3 subscales, which were: perceived risk scale, technology/social influence scale, and anxiety scale. The Cronbach's alpha for these scales were 0.876, 0.858, and 0.827, respectively. Moreover, chatGPT usage was also measured using 4 subscales: perceived usefulness, perceived risk, perceived ease of use, and behaviour scales. The Cronbach's alpha for these scales were: 0.885, 0.718, 0.824, and 0.781, respectively [29].

## Data collection process

The questionnaire was developed using Google Forms and distributed exclusively through formal student communication channels associated with the participants' medical institutions, including platforms such as WhatsApp, Telegram, Facebook, and LinkedIn. This approach ensured accessibility, ease of participation, and adherence to official dissemination protocols. Collaborators with strong academic credentials facilitated its circulation within these authorized networks.

## Data analysis

Data was cleaned and organized using SPSS version 26. Analysis was done using SPSS version 26. Normality tests (Kolmogorov-Smirnov test) were used the assess the scales, which concluded the non-normal distribution of the scales. Frequency tables and descriptive statistics were used to present the socio-demographic data and ChatGPT-related questions. Chi-square test and Pearson correlation were used to compare participants who heard/used ChatGPT and those who didn't across the various socio-demographic factors. Mann-Whitney test and Kruskal-Wallis tests were used to compare ChatGPT attitude scales and subscales, in addition to usage scales and subscales, against different groups of socio-demographic variables. Linear regression analysis was also used for the ChatGPT attitude scale (continuous variable) to explore associations while adjusting for potential confounders, by entering the variables with a P value less than 0.25 from the previous tests into the model. The used level of significance is 0.05.

## Ethical Considerations

Ethical approval was obtained from the Ministry of Health and Social Development, White Nile State, Sudan (Reference No.: 0240106, dated June 10, 2024). The ethical review committee evaluated the study protocol to ensure compliance with national and international guidelines. Participants provided informed consent electronically, with the assurance that their data would remain anonymized and handled confidentially. Data storage and analysis adhered to strict privacy protocols to protect participants' information throughout the study.

## Results

### Socio-demographic characteristics

The study included a total of 1,443 medical students, with more than half being female (65.8%). The median age of the participants was 22 ± 3 years. The majority of students were single (94.4%), and most resided in their family homes (62.0%). Approximately 82% of participants were unemployed. A significant proportion of students were in their 3rd (30.0%) or 4th (19.3%) year of study. Regarding family income, 42.3% of participants reported a monthly income of 300,000 SDGs or more. More than half of the students (949, 65.8%) had heard of ChatGPT, but only 605 students from the total sample (41.9%) had used it prior to the study (**Table 1**).

### Effect of sociodemographic characteristics on hearing about ChatGPT and using it

There was a significant difference across genders regarding hearing and using ChatGPT (p-value <0.001), as males had the highest percentage of hearing about ChatGPT and using it. Marital status also had significant differences across its categories regarding both hearing about ChatGPT and usage (p-values were <0.001 and 0.041, respectively). Single students reported the highest percentage of hearing about ChatGPT and using it. Moreover, hearing about ChatGPT significantly differed across different types of residences during university (p-value = 0.001), as students living in private residences had the highest percentage. ChatGPT usage was significantly different across working statuses (p-value = 0.003), with owning an individual business associating with the highest percentage. Both family income and residency during holidays were significantly different across their respective categories regarding hearing about ChatGPT (p-value <0.001), with the highest percentage among high-income students (300 thousand or more SDGs) and those living outside Sudan. Usage of ChatGPT was also significant with family incomes (p-value <0.001), as students with the highest family income (300 thousand or more SDGs) had the highest usage percentage (**Table 2**).

### Effect of university and internet-related factors on hearing about ChatGPT and using it:

Type of school certificate, type of university, and internet quality significantly differed across their respective categories regarding hearing about ChatGPT (p-value <0.001), with the highest percentages reported among: International General Certificate of Secondary Education students, public university students, and students with excellent internet quality. Internet quality also differed significantly regarding ChatGPT usage (p-value = 0.004), and students with excellent quality had the best usage. Mode of internet access also had a significant difference regarding hearing about ChatGPT (p-value = 0.009), with students using WI-FI internet having the highest percentage of hearing about ChatGPT. The total number of research projects the students have been involved in significantly affected both hearing about ChatGPT and its usage (p-value is 0.001 and <0.001, respectively) (**Table 3**).

### Factors affecting attitude scales toward ChatGPT

The perceived risk scale was statistically significant among categories of: gender, family income, type of university admission, type of university, and internet quality (p-value = 0.002, 0.025, 0.005, 0.001, and 0.001, respectively). The highest perceived risk was reported among: students with an income more than 100 thousand SDGs, students with

**Table 1. Socio-demographic characteristics of medical students from Sudan (N = 1443).**

| Variables | Category | Frequency (percentage) |
|---|---|---|
| Age | | 22 ± 3(Median and IQR) |
| Gender | Male | 494 (34.2%) |
| | Female | 949 (65.8%) |
| Marital Status | Single | 1362 (94.4%) |
| | Engaged | 56 (3.9%) |
| | Married | 24 (1.7%) |
| | Divorced | 1 (.1%) |
| Residency during Studying | Family house | 894 (62.0%) |
| | With relatives | 93 (6.4%) |
| | University Dormitory | 244 (16.9%) |
| | Private Dormitory | 98 (6.8%) |
| | Private residency | 65 (4.5%) |
| | Other | 49 (3.4%) |
| Working status | Freelancer | 117 (8.1%) |
| | Employee | 37 (2.6%) |
| | Own a business | 93 (6.4%) |
| | No work at all | 1196 (82.9%) |
| Academic year | 1st Grade | 153 (10.6%) |
| | 2nd Grade | 269 (18.6%) |
| | 3rd Grade | 433 (30.0%) |
| | 4th Grade | 278 (19.3%) |
| | 5th Grade | 161 (11.2%) |
| | 6th Grade | 149 (10.3%) |
| Average income in Sudanese currency (SDG) | 50,000 or less | 110 (7.6%) |
| | 50,000 - 100,000 | 176 (12.2%) |
| | 100,000 - 200,000 | 264 (18.3%) |
| | 200,000 - 300,000 | 282 (19.5%) |
| | 300,000 or more | 611 (42.3%) |
| Residency during Holidays | Inside Sudan | 1167 (80.9%) |
| | Outside Sudan | 276 (19.1%) |
| Heard of ChatGPT | Yes | 949 (65.8%) |
| | No | 494 (34.2%) |
| Used ChatGPT | Yes | 605 (41.9%) |
| | No | 838 (58.1%) |

a Interquartile range

employed children, public university admission, public university students, and students with an excellent internet connection.

Regarding the technology/social influence scale, it showed significance across categories of: working status, type of university, internet quality, and involvement in extracurricular activities (p-value = 0.012, 0.005, 0.018, and 0.008, respectively). The highest technology/social influence was reported from: students with their businesses, private universities' students, students with a bad internet connection, and students involved in extracurricular activities. Moreover, the number of research projects the student was involved in had a significant, positive, and very low correlation with the technology/social influence scale (r = 0.108, p-value = 0.001).

**Table 2.** Univariate analysis of socio-demographic characteristics of medical students against hearing about ChatGPT and its usage (N = 1443).

| Variables | | Hearing about ChatGPT before the study | | | Usage of ChatGPT before the study | | |
|---|---|---|---|---|---|---|---|
| | | Yes | No | p-value | Yes | No | p-value |
| Age | | 22 (median) | 22 (median) | 0.376 | 22 (median) | 22 (median) | 0.759 |
| gender | Male | 371 (75.1%) | 123 (24.9%) | <0.001* | 273 (55.3%) | 221 (44.7%) | <0.001 |
| | Female | 578 (60.9%) | 371 (39.1%) | | 332 (35.0%) | 617 (65.0%) | |
| Marital Status | Single | 907 (66.6%) | 455 (33.4%) | <0.001 | 578 (42.4%) | 784 (57.6%) | 0.041* |
| | Engaged | 36 (64.3%) | 20 (35.7%) | | 23 (41.1%) | 33 (58.9%) | |
| | Married | 6 (25.0%) | 18 (75.0%) | | 4 (16.7%) | 20 (83.3%) | |
| | Divorced | 0 | 1 (100.0%) | | 0 | 1 (100.0%) | |
| | Widowed | 0 | 0 | | 0 | 0 | |
| Residency during Studying | Family house | 618 (69.1%) | 276 (30.9%) | 0.001* | 378 (42.3%) | 516 (57.7%) | 0.088 |
| | With relatives | 58 (62.4%) | 35 (37.6%) | | 42 (45.2%) | 51 (54.8%) | |
| | University Dormitory | 139 (57.0%) | 105 (43.0%) | | 99 (40.6%) | 145 (59.4%) | |
| | Private Dormitory | 63 (64.3%) | 35 (35.7%) | | 38 (38.8%) | 60 (61.2%) | |
| | Private residency | 47 (72.3%) | 18 (27.7%) | | 35 (53.8%) | 30 (46.2%) | |
| | Other | 24 (49.0%) | 25 (51.0%) | | 13 (26.5%) | 36 (73.5%) | |
| Working status | Freelancer | 82 (70.1%) | 35 (29.9%) | 0.280 | 59 (50.4%) | 58 (49.6%) | 0.003* |
| | Employee | 23 (62.2%) | 14 (37.8%) | | 14 (37.8%) | 23 (62.2%) | |
| | Own a business | 68 (73.1%) | 25 (26.9%) | | 53 (57.0%) | 40 (43.0%) | |
| | No work at all | 776 (64.9%) | 420 (35.1%) | | 479 (40.1%) | 717 (59.9%) | |
| Academic year | 1st Grade | 100 (65.4%) | 53 (34.6%) | 0.331 | 61 (39.9%) | 92 (60.1%) | 0.736 |
| | 2nd Grade | 161 (59.9%) | 108 (40.1%) | | 106 (39.4%) | 163 (60.6%) | |
| | 3rd Grade | 296 (68.4%) | 137 (31.6%) | | 179 (41.3%) | 254 (58.7%) | |
| | 4th Grade | 184 (66.2%) | 94 (33.8%) | | 121 (43.5%) | 157 (56.5%) | |
| | 5th Grade | 108 (67.1%) | 53 (32.9%) | | 75 (46.6%) | 86 (53.4%) | |
| | 6th Grade | 100 (67.1%) | 49 (32.9%) | | 63 (42.3%) | 86 (57.7%) | |
| Average income in Sudanese currency (SDG) | 50,000 or less | 65 (59.1%) | 45 (40.9%) | <0.001* | 43 (39.1%) | 67 (60.9%) | <0.001* |
| | 50,000 - 100,000 | 92 (52.3%) | 84 (47.7%) | | 56 (31.8%) | 120 (68.2%) | |
| | 100,000 - 200,000 | 156 (59.1%) | 108 (40.9%) | | 100 (37.9%) | 164 (62.1%) | |
| | 200,000 - 300,000 | 180 (63.8%) | 102 (36.2%) | | 106 (37.6%) | 176 (62.4%) | |
| | 300,000 or more | 456 (74.6%) | 155 (25.4%) | | 300 (49.1%) | 311 (50.9%) | |
| Residency during holidays | Inside Sudan | 741 (63.5%) | 426 (36.5%) | <0.001* | 479 (41.0%) | 688 (59.0%) | 0.185 |
| | Outside Sudan | 208 (75.4%) | 68 (24.6%) | | 126 (45.7%) | 150 (54.3%) | |

*Statistically significant (p-value <0.05)

Anxiety scales weren't significant with any of the participants' characteristics. However, multiple variables have been found to be statistically significant with the total attitude scale: gender, Type of university admission, and internet quality (p-value = 0.011, 0.023, and 0.028, respectively) (Tables 4, 5).

**Linear regression analysis of factors affecting ChatGPT total attitude Scale**

Linear regression analysis was performed for the ChatGPT Attitude Scale. The model was statistically significant (p-value <0.001) with an R-squire value of 0.031. Gender, type of admission, type of university, and working status were all significant in the model (p-value = 0.010, 0.040, 0.004, and 0.013, respectively) (**Table 6**).

**Table 3. Univariate analysis of university and internet-related factors against hearing about ChatGPT and its usage among medical students (N = 1443).**

| Variables | | Have you heard of ChatGPT before the study? | | | Have you used Chat GPT before the study? | | |
|---|---|---|---|---|---|---|---|
| | | Yes | No | p-value | Yes | No | p-value |
| Type of high school certificate | Sudanese Certificate | 855 (64.9%) | 462 (35.1%) | 0.001* | 548 (41.6%) | 769 (58.4%) | 0.054 |
| | Arabic Countries Certificate | 64 (67.4%) | 31 (32.6%) | | 38 (40.0%) | 57 (60.0%) | |
| | The International General Certificate of Secondary Education (IGCSE) | 29 (96.7%) | 1 (3.3%) | | 19 (63.3%) | 11 (36.7%) | |
| Type of university admission | Public | 668 (67.3%) | 324 (32.7%) | 0.075 | 434 (43.8%) | 558 (56.3%) | 0.095 |
| | Private | 277 (62.5%) | 166 (37.5%) | | 168 (37.9%) | 275 (62.1%) | |
| | Employee children | 3 (42.9%) | 4 (57.1%) | | 2 (28.6%) | 5 (71.4%) | |
| Type of university | Public | 742 (68.4%) | 343 (31.6%) | <0.001* | 470 (43.3%) | 615 (56.7%) | 0.071 |
| | Private | 207 (57.8%) | 151 (42.2%) | | 135 (37.7%) | 223 (62.3%) | |
| Internet quality | Bad | 88 (58.3%) | 63 (41.7%) | <0.001* | 52 (34.4%) | 99 (65.6%) | 0.004* |
| | Average | 396 (69.6%) | 173 (30.4%) | | 241 (42.4%) | 328 (57.6%) | |
| | Good | 259 (58.1%) | 187 (41.9%) | | 173 (38.8%) | 273 (61.2%) | |
| | Excellent | 206 (74.4%) | 71 (25.6%) | | 139 (50.2%) | 138 (49.8%) | |
| Internet Access Mode | Wi-Fi | 167 (74.2%) | 58 (25.8%) | 0.009* | 109 (48.4%) | 116 (51.6%) | 0.098 |
| | Broadband | 5 (71.4%) | 2 (28.6%) | | 3 (42.9%) | 4 (57.1%) | |
| | Mobile internet | 777 (64.2%) | 434 (35.8%) | | 493 (40.7%) | 718 (59.3%) | |
| Perceived academic performance | Failing | 14 (66.7%) | 7 (33.3%) | 0.481 | 8 (38.1%) | 13 (61.9%) | 0.114 |
| | Below average | 82 (71.9%) | 32 (28.1%) | | 60 (52.6%) | 54 (47.4%) | |
| | Average | 554 (64.6%) | 303 (35.4%) | | 350 (40.8%) | 507 (59.2%) | |
| | Above average | 299 (66.3%) | 152 (33.7%) | | 187 (41.5%) | 264 (58.5%) | |
| Engagement In Extracurricular Activity | Yes | 690 (66.7%) | 344 (33.3%) | 0.243 | 443 (42.8%) | 591 (57.2%) | 0.288 |
| | No | 259 (63.3%) | 150 (36.7%) | | 162 (39.6%) | 247 (60.4%) | |
| Number of researches | | 0 (Median) | 0 (Median) | 0.001* | 0 (Median) | 0 (Median) | <0.001* |

*Statistically significant (p-value <0.05)

**Table 4. Univariate analysis of medical students' sociodemographic factors against ChatGPT attitude scales (N = 949).**

| Variables | | Perceived risk Scale | | technology/social influence scale | | Anxiety scale | | Attitude Scale | |
|---|---|---|---|---|---|---|---|---|---|
| | | Median±IQR[a] | p-value | Median±IQR[a] | p-value | Median±IQR[a] | p-value | Median±IQR[a] | p-value |
| Age[b] | | 13±5 | 0.218 (-0.040) | 15±4 | 0.134 (0.049) | 7±4 | 0.084 (0.056) | 34±8 | 0.748 (0.011) |
| Gender | Male | 13±5 | 0.002* | 15±3 | 0.128 | 7±4 | 0.319 | 34±8 | 0.011* |
| | Female | 13±5 | | 14±4 | | 7±4 | | 33±8 | |
| Marital status | Single | 13±5 | 0.830 | 15±4 | 0.328 | 6±4 | 0.081 | 34±8 | 0.662 |
| | Engaged | 12±5 | | 14±3 | | 8±4 | | 33±10 | |
| | Married | 14±3 | | 16.5±5 | | 7±5 | | 38±10 | |
| Working Status | Freelancer | 13±6 | 0.815 | 15±3 | 0.012* | 8±4 | 0.021 | 35±9 | 0.052 |
| | Employee | 13±6 | | 15.5±4 | | 7±5 | | 34±9 | |
| | Own a business | 13±4 | | 16±4 | | 7±3 | | 34±8 | |
| | No work at all | 13±5 | | 14±4 | | 6±4 | | 34±8 | |
| Academic year | 1st Grade | 13±6 | 0.810 | 15±3.5 | 0.307 | 6±3.5 | 0.503 | 35±8 | 0.550 |
| | 2nd Grade | 13±4 | | 14±4 | | 6.5±3.5 | | 34±8 | |
| | 3rd Grade | 13±5 | | 14±4 | | 7±4 | | 34±8 | |
| | 4th Grade | 13±4 | | 14±4 | | 7±4 | | 34±6 | |
| | 5th Grade | 12±5 | | 15±3 | | 6±4 | | 33±9 | |
| | 6th Grade | 13±6 | | 15±3.5 | | 7±5 | | 35±10 | |
| Average income in Sudanese currency (SDG) | 50,000 or less | 12±5 | 0.025* | 14±4 | 0.196 | 7±3 | 0.386 | 33±6 | 0.451 |
| | 50,000 - 100,000 | 12±4 | | 15±4 | | 6±3 | | 34±6 | |
| | 100,000 - 200,000 | 13±4 | | 15±4 | | 7±3 | | 35±7 | |
| | 200,000 - 300,000 | 13±5 | | 15±4 | | 6±4 | | 34±8 | |
| | 300,000 or more | 13±6 | | 14±4 | | 6±4 | | 34±9 | |

*Statistically significant (p-value <0.05)

[a]Interquartile Range

[b]The relationship was calculated using the Spearman correlation test as p-value (r-value).

## Factors affecting ChatGPT usage scales

Working status had significant differences across its categories regarding behaviour scale (p-value = 0.022), with Employees having a higher scale than others. Family income had also significant differences across different income statuses regarding the Perceived usefulness Scale (p-value = 0.020), as students with 100–200 thousand SDGs had the lowest perceived usefulness. Type of admission was found to be significantly related to the Perceived Risk Scale (p-value = 0.014), with public admission students having the highest perceived risk. The type of the university itself also differed significantly regarding all scales of usage except perceived ease of use (p-value <0.05), with public university students having the highest usage scale. Moreover, internet quality differed significantly regarding all scales of usage except perceived ease of use (p-value <0.05), as students with bad connections reported higher usage scale than others. Involvement in extra-curricular activities also significantly affected the perceived usefulness Scale and behaviour Scale (p-value = 0.039 and 0.020, respectively). Age was found to have a significant, very low, and positive correlation with the perceived usefulness scale (p-value = 0.031, r = 0.088) (Tables 7, 8).

## Discussion

This study investigates the awareness and usage of ChatGPT among 1,443 medical students, with a focus on exploring the interplay of various factors like perceived risk, social influence, and anxiety that influence its adoption in medical

**Table 5. Univariate analysis of university and internet-related factors against ChatGPT attitude scales (N = 949).**

| Variables | | Perceived risk Scale | | technology/social influence scale | | Anxiety scale | | Attitude Scale | |
|---|---|---|---|---|---|---|---|---|---|
| | | Median±IQR[a] | p-value | Median±IQR[a] | p-value | Median±IQR[a] | p-value | Median±IQR[a] | p-value |
| Type of certificate | Sudanese Certificate | 13±5 | 0.246 | 15±4 | 0.579 | 7±4 | 0.750 | 34±7 | 0.784 |
| | Arabic Countries Certificate | 13±4.5 | | 15±5 | | 6±4 | | 33±9 | |
| | The International General Certificate of Secondary Education (IGCSE) | 13±5 | | 14±3.5 | | 7±5 | | 33±10.5 | |
| Type of university admission | Public | 13±5 | 0.005* | 14.5±4 | 0.213 | 7±4 | 0.239 | 34±8 | 0.023* |
| | Private | 12±5 | | 15±3 | | 6±3 | | 33±7 | |
| | Employee children | 14±11 | | 11±4 | | 8±4 | | 32±12 | |
| Type of university | Public | 13±5 | 0.001* | 14±4 | 0.005* | 7±4 | 0.207 | 34±8 | 0.087 |
| | Private | 12±5 | | 15±4 | | 6±4 | | 33±7 | |
| Internet quality | Bad | 11±4.5 | 0.001* | 16±4 | 0.018* | 6±3 | 0.115 | 32±7 | 0.028* |
| | Average | 13±4 | | 14±4 | | 6±4 | | 34±8 | |
| | Good | 13±5 | | 15±3 | | 6±4 | | 34±7 | |
| | Excellent | 14±5 | | 15±4 | | 7±4 | | 35±7.5 | |
| Internet Access Mode | Wi-Fi | 12±5 | 0.390 | 14±3 | 0.532 | 6±4 | 0.529 | 33±9 | 0.353 |
| | Broadband | 14±5 | | 14±4 | | 8±2 | | 35±2 | |
| | Mobile internet | 13±5 | | 15±4 | | 7±4 | | 34±8 | |
| Perceived academic performance | Failing | 11±2 | 0.112 | 14±3 | 0.675 | 6.5±3 | 0.854 | 32±6 | 0.439 |
| | Below average | 13±5 | | 14±3 | | 7±4 | | 33.5±5 | |
| | Average | 13±5 | | 15±3 | | 7±4 | | 34±8 | |
| | Above average | 13±5 | | 15±4 | | 7±4 | | 34±9 | |
| Engagement In Extracurricular Activity | Yes | 13±5 | 0.628 | 15±3 | 0.008* | 6±4 | 0.958 | 34±8.5 | 0.480 |
| | No | 13±5 | | 14±4 | | 7±4 | | 34±8 | |
| Number of researches[b] | | 13±5 | 0.071 (-0.059) | 15±4 | 0.001*(0.108) | 7±4 | 0.389 (-0.028) | 34±8 | 0.581 (-0.018) |

*Statistically significant (p-value <0.05)

[a]Interquartile Range

**Table 6. Linear regression analysis of factors affecting ChatGPT total attitude Scale (N = 949).**

| variables | Unstandardized Coefficients | | Standardized Coefficients | t | p-value | 95.0% Confidence Interval for B | | Collinearity Statistics | |
|---|---|---|---|---|---|---|---|---|---|
| | B | Std. Error | Beta | | | Lower Bound | Upper Bound | Tolerance | VIF |
| (Constant) | 38.345 | 1.416 | | 27.075 | .000 | 35.565 | 41.124 | | |
| Gender | -1.135 | .438 | -.085 | -2.592 | .010* | -1.995 | -.276 | .975 | 1.026 |
| Type of university admission | -1.094 | .533 | -.078 | -2.053 | .040* | -2.141 | -.048 | .728 | 1.374 |
| Internet quality | .657 | .228 | .093 | 2.880 | .004* | .209 | 1.105 | .997 | 1.003 |
| Type of university | -.102 | .599 | -.006 | -.170 | .865 | -1.277 | 1.073 | .723 | 1.383 |
| Working status | -.597 | .240 | -.082 | -2.487 | .013* | -1.068 | -.126 | .971 | 1.030 |

Dependent Variable: Attitude Scale

*Statistically significant (p-value <0.05)

**Table 7. Univariate analysis of medical students' characteristics against ChatGPT usage scales (N = 605).**

| variables | | Perceived usefulness Scale | | Perceived risk Scale | | Perceived ease of use Scale | | Behavior Scale | | Usage Scale | |
|---|---|---|---|---|---|---|---|---|---|---|---|
| | | Median±IQR[a] | p-value | Median±IQR[a] | p-value | Median±IQR[a] | p-value | Median±IQR[a] | p-value | Median±IQR[a] | p-value |
| Age[b] | | 22±5 | 0.031* (0.088) | 8±3 | 0.117 (-0.064) | 8±2 | 0.710 (-0.015) | 9±5 | 0.810 (-0.010) | 48±10 | 0.606 (0.021) |
| Gender | Male | 22±6 | 0.360 | 8±4 | 0.078 | 8±2 | 0.069 | 9±5 | 0.747 | 48±10 | 0.724 |
| | Female | 22±6 | | 7±3 | | 8±2 | | 9±4 | | 48±9 | |
| Marital status | Single | 22±5 | 0.171 | 8±3 | 0.753 | 8±2 | 0.504 | 9±5 | 0.509 | 48±10 | 0.074 |
| | Engaged | 22±6 | | 8±3 | | 8±4 | | 9±5 | | 48±13 | |
| | Married | 25±4.5 | | 7±3 | | 9±2 | | 14.5±3.5 | | 54.5±5 | |
| Working status | Free-lancer | 23±6 | 0.635 | 8±4 | 0.435 | 8±3 | 0.185 | 10±4 | 0.022* | 50±9 | 0.176 |
| | Employee | 22.5±5 | | 8±3 | | 9.5±2 | | 11±4 | | 51.5±14 | |
| | Own a business | 22±7 | | 7±3 | | 8±3 | | 10±5 | | 47±13 | |
| | No work at all | 22±5 | | 8±3 | | 8±2 | | 9±4 | | 47±9 | |
| Academic year | 1st Grade | 22±5 | 0.671 | 8±4 | 0.615 | 8±2 | 0.962 | 9±3 | 0.070 | 48±11 | 0.710 |
| | 2nd Grade | 21±5 | | 8±3 | | 8±2 | | 9±5 | | 47±8 | |
| | 3rd Grade | 22±6 | | 8±3 | | 8±2 | | 10±4 | | 49±11 | |
| | 4th Grade | 22±5 | | 7±3 | | 8±2 | | 10±5 | | 48±10 | |
| | 5th Grade | 22±5 | | 7±3 | | 8±2 | | 9±5 | | 48±9 | |
| | 6th Grade | 22±7 | | 8±3 | | 8±2 | | 8±4 | | 47±8 | |
| Average income in Sudanese currency (SDG) | 50,000 or less | 23±5 | 0.020* | 7±3 | 0.052 | 8±3 | 0.072 | 10±4 | 0.013 | 50±9 | 0.108 |
| | 50,000 - 100,000 | 23±6 | | 7±3 | | 8±2 | | 10±3 | | 48±7.5 | |
| | 100,000 - 200,000 | 21.5±6 | | 8±2.5 | | 8±2.5 | | 9±3.5 | | 47±9 | |
| | 200,000 - 300,000 | 23.5±7 | | 7±3 | | 9±2 | | 10±4 | | 49±9 | |
| | 300,000 or more | 22±5 | | 8±3 | | 9±2 | | 9±5 | | 47±10 | |

*Statistically significant (p-value <0.05)

[a]Interquartile Range

[b]The relationship was calculated using the Spearman correlation test as p value (r value).

education. By analyzing the participants' socio-demographic characteristics, academic backgrounds, and modes of internet access, the research provides nuanced insights into how these elements converge to shape the integration of artificial intelligence (AI) tools like ChatGPT in the learning journey of future healthcare professionals.

In our study, nearly one-third of participants (34.2%) were entirely unaware of ChatGPT, while the majority (65.8%) had a previous encounter. However, only 41.9% of the total sample indicated that they had actually used it., highlighting a notable gap between awareness and practical application. These findings align with similar studies conducted in Malaysia and India [30,31]. This gap can be due to lack of prior knowledge on effective utilization, as some might be aware of the presence of ChatGPT but has no background about its real capabilities [32], also ethical concerns and privacy issues can be some of the influencing factors, as Ganjavi has reported that students have expressed their concerns regarding

**Table 8. Univariate analysis of university and internet-related factors against ChatGPT usage scales (N = 605).**

| variables | | Perceived usefulness Scale | | Perceived risk Scale | | Perceived ease of use Scale | | Behavior Scale | | Usage Scale | |
|---|---|---|---|---|---|---|---|---|---|---|---|
| | | Median±IQR[a] | p-value | Median±IQR[a] | p-value | Median±IQR[a] | p-value | Median±IQR[a] | p-value | Median±IQR[a] | p-value |
| Type of certificate | Sudanese Certificate | 22±5 | 0.116 | 8±3 | 0.585 | 8±2 | 0.167 | 9±5 | 0.452 | 48±10 | 0.348 |
| | Arabic Countries Certificate | 23.5±6 | | 7.5±3 | | 10±2 | | 10±4 | | 49±10 | |
| | The International General Certificate of Secondary Education (IGCSE) | 20±7 | | 7±4 | | 8±2 | | 8±6 | | 46±13 | |
| Type of university admission | Public | 22±5 | 0.429 | 8±3 | 0.014* | 8±2 | 0.913 | 9±5 | 0.123 | 48±10 | 0.930 |
| | Private | 22±6 | | 7±3 | | 8±2 | | 10±4 | | 47.5±10 | |
| | Employee children | 24±8 | | 6±6 | | 8.5±1 | | 10±4 | | 48.5±19 | |
| Type of university | Public | 22±5 | 0.003* | 8±3 | 0.007* | 8±2 | 0.202 | 9±4 | <0.001* | 47±10 | 0.027* |
| | Private | 23±5 | | 7±3 | | 8±2 | | 10±4 | | 49±9 | |
| Internet quality | Bad | 23±5 | 0.032* | 7±2.5 | 0.033* | 9±2 | 0.218 | 12±4 | <0.001* | 50±9 | 0.018* |
| | Average | 21±6 | | 8±3 | | 8±2 | | 9±4 | | 46±9 | |
| | Good | 23±6 | | 8±3 | | 8±2 | | 10±4 | | 49±9 | |
| | Excellent | 22±5 | | 8±4 | | 9±2 | | 9±5 | | 47±10 | |
| Internet Access Mode | Wi-Fi | 23±7 | 0.516 | 7±3 | 0.737 | 9±2 | 0.662 | 10±4 | 0.013 | 49±11 | 0.237 |
| | Broadband | 17±9 | | 8±1 | | 8±6 | | 6±7 | | 41±19 | |
| | Mobile internet | 22±5 | | 8±3 | | 8±2 | | 9±4 | | 47±10 | |
| Perceived academic performance | Failing | 23±5 | 0.653 | 6.5±1.5 | 0.366 | 8±0.5 | 0.230 | 11.5±4 | 0.262 | 51±11 | 0.284 |
| | Below average | 21±7.5 | | 7.5±3 | | 8±3 | | 8±5 | | 46±11 | |
| | Average | 22±5 | | 8±3 | | 8±2 | | 9±4 | | 48±10 | |
| | Above average | 22±5 | | 8±3 | | 9±2 | | 9±5 | | 48±8 | |
| Engagement In Extracurricular Activity | Yes | 22±6 | 0.039* | 7±3 | 0.129 | 8±2 | 0.237 | 10±4 | 0.020* | 48±9 | 0.075 |
| | No | 22±6 | | 8±3 | | 9±2 | | 9±4 | | 47±11 | |
| Number of researches[b] | | 22±5 | 0.220 | 8±3 | 0.193 | 8±2 | 0.908 | 9±5 | 0.559 | 48±10 | 0.739 |

*Statistically significant (p-value <0.05)

[a]Interquartile Range

plagiarism, patient privacy, and the inaccurate responses that chat provided [33]. Besides the absence of educational interventions that can introduce the student to the capabilities and limitations of ChatGPT [34].

Multiple factors were found to significantly influence ChatGPT attitude: gender, university admission type, and internet quality (p = 0.011, 0.023, and 0.028, respectively). Male students, private university students, and those with better internet access demonstrated more favorable attitudes, reinforcing the importance of socio-demographic and infrastructural factors in shaping AI perceptions [35].

**Socio-demographic factors: Can Socio-demographic factors predict who knows and uses ChatGPT?**

This research uncovered important gender-related disparities in ChatGPT awareness and usage, as male participants demonstrated considerably greater familiarity (75.1%) and usage rates (55.3%) than females (60.9% and 35.0%, respectively; p < 0.001), which aling with a study in UAE that reported significantly higher usage among males [36], apart of UAE's study, these results challenge earlier research, such as studies by Faouzi Kamoun et al. (2023) and Hu et al. (2023), which reported no notable gender disparities in technology adoption [37,38]. Interestingly, a Malaysian study by Pallivathukal et al. (2024) found the opposite trend, where females demonstrated greater knowledge and usage of AI tools compared to males [30]. However, in this study, males consistently outperformed females, highlighting a different dynamic that may reflect unique cultural or societal influences.

In the Sudanese context, these gender differences may be rooted in broader societal and cultural norms. Traditional settings often afford men greater exposure to technology while imposing domestic responsibilities and resource limitations on women [39]. These limitations might impede women's chances to interact with and embrace new technologies such as ChatGPT. These results highlight the urgent requirement for gender-equity programs that tackle these systemic obstacles. Customized educational initiatives, enhanced resource availability, and focused digital literacy outreach are vital to enable women and foster inclusivity in AI adoption, ultimately narrowing the gender divide in tech involvement.

Marital status proved to be a key factor affecting ChatGPT adoption, as singles showed significantly greater levels of awareness (66.6%) and usage (42.4%) than their married counterparts (25.0% and 16.7%; p < 0.001 and p = 0.041, respectively). Although the precise causes of these differences are not fully understood, it is suggested that married people might encounter conflicting family obligations that restrict their capacity to use non-essential technologies, besides the possibility of having children which was previously found to negatively affect ChatGPT activity in Germany [40]. On the other hand, singles, having less personal commitments, probably enjoy increased flexibility and freedom to experiment with innovative tools such as ChatGPT. More research is required to gain a clearer insight into these dynamics and the wider impacts on technology adoption across various demographic groups.

Living arrangements were found to have a significant impact on ChatGPT engagement, with students living in private accommodations reporting the highest levels of awareness (72.3%; p = 0.001). The independence and access to private resources in these settings likely encourage the exploration and use of digital tools like ChatGPT. It is worth mentioning the probability of its usage for companionship and emotional support, based on its ability to generate humanized conversations to compensate for loneliness and boredom [41]. In contrast, students residing in dormitories or family homes often encounter shared resources and infrastructural challenges, such as limited internet access, which can hinder their ability to fully engage with technology. Additionally, students involved in entrepreneurial activities reported the highest ChatGPT usage (57.0%; p = 0.003), emphasizing the practical value of AI tools in business contexts. This finding aligns with McKinsey's company projection that ChatGPT has the potential to significantly transform entrepreneurial activities [42]. A study in Germany reported that full time employment was one of the factors that negatively affected the ChatGPT activity among the participants [40], this can be due to the stressful nature of work for the employees compared to business owners, since the latter requires critical thinking and a lot of analysis to take decisions, which ChatGPT can help them with, compared to employees who suffer from work pressure, in addition to the availability of internal resources for the employees, which reduces their need to search the Internet or engaging with ChatGPT.

Given these insights, universities should focus on resource allocation strategies that improve access to digital tools, especially for students in less conducive living environments, to promote digital innovation and equitable technological engagement.

Economic disparities were a key factor influencing ChatGPT engagement, with students from families earning over 300,000 SDGs showing the highest levels of awareness and usage (p < 0.001). High-income households typically offer better access to reliable devices and internet connections, which in turn facilitates greater technological engagement [43], these findings were consistent with previous studies shown that students from low socio-economic backgrounds face

difficulties in accessing high internet speed, reliable digital devices, and online resources which hinder the improvement of their digital skills and their technological engagement [44]. Moreover, students with global experience indicated greater familiarity with ChatGPT, highlighting the influence of engagement with advanced technological environments. These results emphasize the persistent digital gap among socio-economic groups and underline the pressing necessity for policies and infrastructure enhancements that guarantee fair access to technological resources. Tackling these inequalities is crucial for promoting inclusive digital literacy and AI integration.

### Academic background and institutional impact: How do educational curricula and research engagement shape ChatGPT adoption, and in turn, how does ChatGPT adoption influence these factors among Sudanese medical students?

Our analysis revealed that high school certification served as a major indicator of ChatGPT awareness and utilization ($p < 0.001$), with students from internationally acknowledged programs, like the International General Certificate of Secondary Education (IGCSE), showing the greatest levels of awareness (96.7%) and usage (63.3%) [45]. Conversely, learners from conventional Sudanese or Arabic educational frameworks indicated significantly reduced levels of engagement, emphasizing how educational background influences familiarity with digital technologies and AI applications such as ChatGPT. This discovery highlights the significance of incorporating contemporary technological education into various curricula to guarantee wider access to and comprehension of new technologies [46]. These findings indicate that international curricula strongly prioritize digital literacy, providing students with the abilities needed to embrace and proficiently utilize AI tools. This underscores the necessity for curriculum changes in conventional education models to more effectively equip students for the challenges of the digital era. Incorporating digital literacy into traditional education can help close the divide in technological involvement and guarantee that students are properly prepared to manage and succeed in a progressively digital environment.

Research engagement was significantly associated with ChatGPT utilization ($p < 0.001$), as students participating in research exhibited greater awareness and use. This is likely due to their familiarity with academic technologies and tools. The question arises whether engaging in research drives AI adoption or if exposure to AI tools enhances research activities. Either way, these findings underscore the significant role AI can play in boosting academic productivity, especially in areas like data analysis and literature reviews. To ensure students are well-equipped for the future, universities should prioritize integrating AI literacy into research training, helping students develop the technological skills necessary for success in academia [47].

Additionally, students from public universities reported higher levels of ChatGPT awareness (68.4%) compared to those from private universities (57.8%; $p < 0.001$), although usage rates were similar between the two groups ($p = 0.071$). This disparity may be attributed to the research-oriented focus of public institutions, which often encourage engagement with emerging technologies, in contrast to the service-oriented emphasis of private universities [48].To bridge this gap, it is essential to integrate AI tools into the curricula of all types of universities, ensuring that students from both public and private institutions are equally prepared for the technological landscape [49].

### Digital infrastructure: How does internet quality impact students' ability to engage with and benefit from AI technologies like ChatGPT?

Internet quality emerged as a critical factor influencing ChatGPT engagement among students. Those with excellent internet access reported the highest levels of awareness (74.4%) and usage (50.2%), while students with poor internet access demonstrated significantly lower levels of engagement ($p < 0.001$ and $p = 0.004$). Interestingly, students who relied on Wi-Fi networks exhibited higher awareness (74.2%) compared to those using mobile internet, indicating that stable and reliable internet connections are essential for effective use of AI technologies, as it can increase the perceived ease of use among the students and therefore there overall engagement with Chat GPT as previous studies have shown [29,36].

These findings underline the importance of internet infrastructure in enabling access to digital tools. In areas with poor or inconsistent internet connectivity, students encounter major obstacles in accessing platforms such as ChatGPT, potentially exacerbating the digital divide. To foster fair participation in the digital era, it is crucial to invest in dependable and accessible internet infrastructure, especially in underserved and rural regions, enabling all students to interact with and take advantage of new technologies such as AI [50].

**Do perceived risk and social influence play a role in ChatGPT adoption among medical students?**

In this study, various factors were found to significantly influence ChatGPT adoption, particularly in relation to perceived risk, social influence, and attitudes toward the technology.

The perceived risk scale indicated notable correlations with various socio-demographic and infrastructural factors, such as gender, family income, type of university, type of admission, and internet quality ($p = 0.002$, $0.025$, $0.005$, $0.001$, and $0.001$, respectively). Students from families with higher incomes, those enjoying reliable internet access, and those enrolled in public universities were more likely to indicate greater levels of perceived risk. This contradicts previous studies, as both Qamar et al and Sallam et al reported no statistically significant differences were found regarding gender, and type of university [29,51], and might reflect the cultural influence and the imbalance between public and private universities in the Sudanese educational system. This may indicate that these groups possess a more critical or nuanced comprehension of the societal effects of AI, especially regarding privacy, data security, and the larger ethical issues that arise with its application. These results emphasize the necessity of tackling the perceived dangers of AI adoption, especially for those who might be more conscious of the wider implications of technology usage.

The social influence scale, which assesses how peer and societal expectations affect technology adoption, demonstrated significant relationships with employment status, university type, internet quality, and extracurricular participation ($p = 0.012$, $0.005$, $0.018$, and $0.008$, respectively). Interestingly, students who were employed, attended private colleges, experienced limited internet access, or participated in extracurricular activities noted a greater impact from their social networks in using ChatGPT. This indicates that learners in these categories are more prone to be swayed by friends, relatives, or social norms when choosing to adopt new technologies. Studies by Strzelecki et al. [52], Bahador et al. [53], and Foroughi et al. [54] reported the significance of social influence on the likelihood of using ChatGPT and other technologies, as students who perceived its usage to be ethical and socially acceptable were more likely to use them. The discovery that students lacking adequate internet access indicated greater social influence is especially fascinating, as it implies that without dependable technological support, people might rely on their social circles for advice on interacting with digital resources [55]. This corresponds with recognized theories of technology adoption, highlighting the significance of social factors, like peer pressure and societal movements, in influencing behavioral intentions [56,57].

Unexpectedly, anxiety showed no significant correlation with the adoption of ChatGPT, which was contrary to initial expectations [56]. However, a study conducted in the UK and Nepal also found that anxiety had no significant effect on the Chat GPT usage among the students [58]. Although anxiety is frequently mentioned as a hindrance to utilizing technology, particularly with complex or unfamiliar tools, our research indicates that regular interaction with AI tools in educational environments might help reduce these apprehensions. It can also be due to the influence of other factors like the perceived usefulness and the peers' encouragement [58]. This might indicate an increasing comfort and familiarity with AI technologies among students, likely diminishing the concerns that may have been more common during the earlier phases of AI integration. Furthermore, the heightened utilization of AI tools in educational settings might foster increased confidence and a more casual attitude towards embracing technology.

Ultimately, the general perception of ChatGPT was affected by variables like gender, university admission type, and internet quality ($p = 0.011$, $0.023$, and $0.028$, respectively). These factors influenced how students view and engage with AI technologies, emphasizing that socio-demographic and infrastructural elements are vital for comprehending technology adoption. The analysis of linear regression showed that gender, type of university admission, university category, and

employment status were all important predictors of students' perceptions of ChatGPT (p = 0.010, 0.040, 0.004, and 0.013, respectively). This emphasizes the intricacy of AI adoption and stresses the necessity for universities to customize interventions that take these different factors into account, encouraging equitable access to and adoption of AI technologies among varied student groups.

## Conclusion

Research emphasizes the complex interaction of socio-demographic elements, educational backgrounds, internet quality, and peer effects in shaping students' perceptions of ChatGPT. Although perceived risks and social influence were identified as key factors in adoption, anxiety seems to be a smaller obstacle than initially expected. The results highlight the significance of confronting systemic inequalities, especially regarding socio-demographic differences, perceived risks, and the effects of social influence.

## Implications

Moving forward, educational institutions must take a comprehensive, evidence-based approach to integrate AI tools like ChatGPT into their curricula. This includes fostering digital literacy and ensuring equitable access to technology, particularly by improving digital infrastructure in underserved areas. To bridge the urban-rural divide, it is crucial to provide students with the resources they need to engage with digital tools effectively. Targeted programs to address gender disparities, such as scholarships and mentorship initiatives for female students, will empower underrepresented groups and encourage diversity in AI adoption. Additionally, institutions should focus on reforming curricula to include AI literacy and practical training, which will better prepare students for the evolving healthcare landscape. Institutional readiness, supported by faculty training and standardized policies, is also key to the sustainable implementation of AI tools. Collectively, these efforts aim to democratize AI access, enhance digital competence, and equip students to leverage transformative technologies, ultimately fostering a more inclusive and innovative academic and professional environment.

## Strengths

A major strength of this study lies in its inclusion of diverse public and private universities, offering a comprehensive dataset that captures the experiences of medical students across Sudan. The large sample size not only exceeds the minimum required for statistical power but also enhances the reliability and generalizability of the results. The use of a validated questionnaire with strong internal consistency (Cronbach's alpha > 0.76) ensures data accuracy, while advanced statistical analyses add credibility to the findings. Moreover, the study's ethical rigor, such as anonymized responses and official approval, strengthens its adherence to research standards and reinforces its reliability.

## Limitations

However, the study's reliance on convenience sampling and digital distribution limits its generalizability, particularly as students from rural or low-resource settings may be underrepresented. The self-reported nature of the data introduces the potential for social desirability bias, and the cross-sectional design and convenience sampling restrict the ability to assess changes over time or causality. Additionally, while the questionnaire's validation ensures reliability, the lack of contextual adaptation to Sudan's unique educational and cultural landscape may affect the depth of the analysis. Moreover, the study's narrow focus on ChatGPT, without a broader exploration of institutional or technological factors, limits its capacity to situate the findings within the wider context of AI adoption in medical education and hinders its applicability to other AI technologies and educational chatbots.

In summary, while this study provides valuable insights into ChatGPT adoption, addressing these limitations in future research will improve its representativeness, depth, and applicability, offering a more comprehensive understanding of the factors influencing AI integration in educational settings.

## Supporting information

**S1 Data.   Raw, anonymized dataset used for the statistical analysis in the study.**
(SAV)

## Acknowledgments

The authors of this study extend their heartfelt gratitude and best regards to Naba Hashim Shatta Salim for her role as a data collector. Special thanks are also extended to Dr. Doaa Abdulrahman Osman Khalid, Mowada Mohamedadam Ali Ibrahim, Alsiddig Gorashi Ibrahim Yasein, and Razaz Muhmoud Hamad Abdelrahman for their valuable contributions to this study.

**Consent for publication:** Availability of data and materials: all data and materials of this study are accessible and available by the corresponding author of this study upon a reasonable request

## Author contributions

**Conceptualization:** Weam Mohamed Meargni Ahmed, Malaz M. Abdalmotalib, Galia Tajelsir Fadulelmula Mohammed, Waad Mohamed Ibrahim Mohamed, Fatima Salih Babiker Mohammed, Hajar Saad Salih, Hiba Omer Yousif Mohamed.

**Data curation:** Weam Mohamed Meargni Ahmed, Malaz M. Abdalmotalib, Galia Tajelsir Fadulelmula Mohammed, Waad Mohamed Ibrahim Mohamed, Fatima Salih Babiker Mohammed, Hajar Saad Salih, Hiba Omer Yousif Mohamed.

**Formal analysis:** Mohamed H. Elbadawi.

**Investigation:** Weam Mohamed Meargni Ahmed, Malaz M. Abdalmotalib, Galia Tajelsir Fadulelmula Mohammed.

**Methodology:** Weam Mohamed Meargni Ahmed, Malaz M. Abdalmotalib, Galia Tajelsir Fadulelmula Mohammed.

**Project administration:** Weam Mohamed Meargni Ahmed, Malaz M. Abdalmotalib.

**Resources:** Weam Mohamed Meargni Ahmed.

**Software:** Malaz M. Abdalmotalib, Mohamed H. Elbadawi.

**Supervision:** Malaz M. Abdalmotalib.

**Validation:** Weam Mohamed Meargni Ahmed.

**Visualization:** Weam Mohamed Meargni Ahmed, Mohamed H. Elbadawi, Galia Tajelsir Fadulelmula Mohammed.

**Writing – original draft:** Weam Mohamed Meargni Ahmed, Malaz M. Abdalmotalib, Mohamed H. Elbadawi, Galia Tajelsir Fadulelmula Mohammed, Waad Mohamed Ibrahim Mohamed, Fatima Salih Babiker Mohammed, Hajar Saad Salih, Hiba Omer Yousif Mohamed.

**Writing – review & editing:** Weam Mohamed Meargni Ahmed, Malaz M. Abdalmotalib, Mohamed H. Elbadawi, Galia Tajelsir Fadulelmula Mohammed, Waad Mohamed Ibrahim Mohamed, Fatima Salih Babiker Mohammed, Hajar Saad Salih, Hiba Omer Yousif Mohamed.

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
