## [Decision Letter · Decision Letter 0]

7 Mar 2025

PONE-D-25-05850Shaping the Future of Medical Education: A Cross-Sectional Study on ChatGPT Attitude and Usage Among Medical Students in Sudan, 2024PLOS ONE

Dear Dr. Abdalmotalib,

Thank you for submitting your manuscript to PLOS ONE. After careful consideration, we feel that it has merit but does not fully meet PLOS ONE’s publication criteria as it currently stands. Therefore, we invite you to submit a revised version of the manuscript that addresses the points raised during the review process.

We look forward to receiving your revised manuscript.

Kind regards,

Sujita Kumar Kar

Academic Editor

PLOS ONE

**Journal Requirements:**

1. When submitting your revision, we need you to address these additional requirements. Please ensure that your manuscript meets PLOS ONE's style requirements, including those for file naming. The PLOS ONE style templates can be found at https://journals.plos.org/plosone/s/file?id=wjVg/PLOSOne_formatting_sample_main_body.pdf and https://journals.plos.org/plosone/s/file?id=ba62/PLOSOne_formatting_sample_title_authors_affiliations.pdf 2. We note that your Data Availability Statement is currently as follows: All relevant data are within the manuscript and its Supporting Information files. Please confirm at this time whether or not your submission contains all raw data required to replicate the results of your study. Authors must share the “minimal data set” for their submission. PLOS defines the minimal data set to consist of the data required to replicate all study findings reported in the article, as well as related metadata and methods (https://journals.plos.org/plosone/s/data-availability#loc-minimal-data-set-definition). For example, authors should submit the following data: - The values behind the means, standard deviations and other measures reported;- The values used to build graphs;- The points extracted from images for analysis. Authors do not need to submit their entire data set if only a portion of the data was used in the reported study. If your submission does not contain these data, please either upload them as Supporting Information files or deposit them to a stable, public repository and provide us with the relevant URLs, DOIs, or accession numbers. For a list of recommended repositories, please see https://journals.plos.org/plosone/s/recommended-repositories. If there are ethical or legal restrictions on sharing a de-identified data set, please explain them in detail (e.g., data contain potentially sensitive information, data are owned by a third-party organization, etc.) and who has imposed them (e.g., an ethics committee). Please also provide contact information for a data access committee, ethics committee, or other institutional body to which data requests may be sent. If data are owned by a third party, please indicate how others may request data access. 3. When completing the data availability statement of the submission form, you indicated that you will make your data available on acceptance. We strongly recommend all authors decide on a data sharing plan before acceptance, as the process can be lengthy and hold up publication timelines. Please note that, though access restrictions are acceptable now, your entire data will need to be made freely accessible if your manuscript is accepted for publication. This policy applies to all data except where public deposition would breach compliance with the protocol approved by your research ethics board. If you are unable to adhere to our open data policy, please kindly revise your statement to explain your reasoning and we will seek the editor's input on an exemption. Please be assured that, once you have provided your new statement, the assessment of your exemption will not hold up the peer review process. 4. Your ethics statement should only appear in the Methods section of your manuscript. If your ethics statement is written in any section besides the Methods, please move it to the Methods section and delete it from any other section. Please ensure that your ethics statement is included in your manuscript, as the ethics statement entered into the online submission form will not be published alongside your manuscript.

**Additional Editor Comments:**

There are some major concerns raised by the reviewer 1 and 3. Kindly address them in detail. Considering the importance of the topic, we are giving a chance to the authors to revise the article.

Reviewers' comments:

Reviewer's Responses to Questions

**Comments to the Author**

1. Is the manuscript technically sound, and do the data support the conclusions?

Reviewer #1: Partly

Reviewer #2: Yes

Reviewer #3: Partly

2. Has the statistical analysis been performed appropriately and rigorously? 

Reviewer #1: Yes

Reviewer #2: Yes

Reviewer #3: Yes

3. Have the authors made all data underlying the findings in their manuscript fully available?

Reviewer #1: Yes

Reviewer #2: Yes

Reviewer #3: Yes

4. Is the manuscript presented in an intelligible fashion and written in standard English?

Reviewer #1: Yes

Reviewer #2: Yes

Reviewer #3: Yes

5. Review Comments to the Author

**Reviewer #1:**  I must congratulate whole team for their excellent effort to put this issue in a setting with a limited resources touching upon the complex issue of demographic profiles. The issue narrates the local issue very relevantly and appears pertinent for their geographical are and definitely is issue for the policy makers so that in an era where AI is going to replace the people who do not use the AI.

The limitations in use is definitely marred by the variety of demographic and societal issues including the financial constraints. These issues have been brought by many in the literature. These issues might not be of added value to the medical students and scientist but definitely of immense help to the social engineers, policy makers and government agencies.

Also for the medical curriculum a study that explores the outcome of AI tools in the performance and enhancement of their performance will be a great help so that it can be an integral part of the curriculum.

**Reviewer #2: ** This study presents an original investigation into the awareness, attitudes, and usage of ChatGPT among Sudanese medical students using a cross-sectional survey design. The manuscript benefits from a large sample size and a clear focus on a contemporary issue in medical education. However, several concerns regarding the conceptual framing, methodology, and reporting must be addressed.

Major Comments

1. Research Gap and Objectives

The introduction does not clearly articulate the specific research gap. While the manuscript discusses AI’s potential in medical education broadly, it fails to specify what prior studies have overlooked, particularly in the context of Sudan. Revise the introduction to explicitly define the research gap and justify the need for this study.

2. Claims and Supporting Evidence

The manuscript describes ChatGPT as a transformative force in medical education and highlights its broad applicability (e.g., clinical decision support) without sufficient empirical evidence or rigorous literature support. Distinguish between potential benefits and established outcomes.

3. Methodological Design

The use of a cross-sectional design limits causal inferences. Additionally, reliance on convenience sampling may introduce selection bias, potentially over-representing students with better digital access. Acknowledge these limitations explicitly in the manuscript. Consider suggesting future research with longitudinal or randomized sampling designs. The rationale behind the use of non-parametric tests and linear regression analysis should be explicitly stated.

4. Data Collection and Instrument Validation

The survey was distributed exclusively through online channels. This digital-only approach risks excluding students with poor or inconsistent internet access. Moreover, while the questionnaire was adapted from previous work, there is no mention of local reliability or validity testing (e.g., Cronbach’s alpha). Authors should provide details on any pilot testing, reliability, and validity analyses performed for the adapted questionnaire. If such analyses have not been conducted, consider including them or discussing this as a limitation.

5. Discussion and Interpretation of Findings

The discussion presents results without connecting them to prior studies.

The discussion section presents the findings largely in isolation, without a cohesive narrative that integrates methodological limitations, contextual factors (e.g., socio-economic and infrastructural challenges in Sudan), and actionable recommendations. The authors need to enhance the discussion by contextualizing your findings within existing literature. Explore the implications of observed trends (such as gender differences) in more depth, and propose concrete strategies for addressing identified challenges in medical education and digital access.

Minor Comments

1.Ensure consistency in terminology when referring to ChatGPT and other AI tools.

2.Clarify any abbreviations (e.g., SDGs for Sudanese currency) when first introduced.

3.Review the overall structure to maintain a logical flow from the introduction through to the conclusions.

The study addresses an important and timely topic in medical education. However, before further consideration, the manuscript requires revisions to clearly define its research gap, strengthen methodological reporting, and provide a more integrated discussion of the findings. Addressing these points will enhance the scientific rigor and clarity of your work.

**Reviewer #3:**  The paper is well-written and provides valuable insights into the awareness, usage, and identification of economical and socio-demographic indicators related to the use of large language models, particularly ChatGPT. We offer several comments to enhance this work further.

1. The introduction lacks adequate citations.

2. The authors statement “In countries like Sudan, where disparities in access to advanced technology are profound, the adoption of AI tools such as ChatGPT is shaped not only by technological factors but also by socio-economic conditions, geographic location, and institutional resources.”, connotes a conclusion without citation. Moreso, these issues are what the authors have indicated to investigate among medical student in Sudan. A critical look should be given to this point.

3. The study failed to offer explicit justifications for selecting ChatGPT as the large language model (LLM) of interest, despite acknowledging its use in medical contexts. Specifically, it did not address why alternatives like Glass Health or Med-PaLM, which are tailored for healthcare applications, were not considered. Furthermore, while large language models (LLM) are widely integrated with major search engines and web browsers such as Google and Brave, the study did not elaborate on the choice of ChatGPT.

4. The equations should be rewritten and cited in the paper.

5. Tables should be redrawn, and figures could be used to improve presentation results.

6. In Table 1, participants were asked about their interaction with ChatGPT. The data reveals that 838 individuals reported using ChatGPT, while only 494 respondents indicated they had not heard of it at all. This significant difference needs clarification to ensure the accuracy and completeness of the information presented.

6. PLOS authors have the option to publish the peer review history of their article (what does this mean? ). If published, this will include your full peer review and any attached files.

**Do you want your identity to be public for this peer review?** For information about this choice, including consent withdrawal, please see our Privacy Policy .

Reviewer #1: **Yes: ** RAJIV GARG

Reviewer #2: **Yes: ** Varchasvi Mudgal

Reviewer #3: No

---

## [Author Response · Author response to Decision Letter 1]

19 Apr 2025

PONE-D-25-05850

Shaping the Future of Medical Education: A Cross-Sectional Study on ChatGPT Attitude and Usage Among Medical Students in Sudan, 2024

Journal Requirements:

Dear editor,

Thank you for your guidance. We have reviewed the PLOS ONE formatting requirements and will ensure that our manuscript fully adheres to the specified style, including file naming conventions.

Dear Editor,

Thank you for your comment and for outlining the data sharing requirements. We confirm that our submission includes all raw data required to replicate the results of our study, in accordance with PLOS ONE’s definition of the minimal data set. This includes the values underlying all reported means, standard deviations, statistical analyses, and figures.

To ensure transparency and compliance with PLOS ONE’s policies, we will upload the SPSS file containing the full raw dataset used in our analysis as a Supporting Information file. This dataset includes all relevant values and metadata necessary to reproduce our findings.

Dear editor,

Thank you for your clarification regarding the data availability policy. We confirm that we will make the complete dataset freely accessible upon acceptance, in full compliance with PLOS ONE’s open data policy. The dataset, which includes all raw values necessary to replicate the study findings, will be uploaded as a Supporting Information file with the manuscript.

We appreciate the importance of transparency and reproducibility, and we do not anticipate any ethical or legal restrictions on data sharing.

Dear editor,

Thank you for your guidance. We have revised the manuscript to ensure that the ethics statement appears only in the Methods section, and we have removed it from all other sections. The statement now clearly describes the ethical approval obtained and the consent procedures followed, following PLOS ONE’s requirements.

Additional Editor Comments:

There are some major concerns raised by the reviewer 1 and 3. Kindly address them in detail. Considering the importance of the topic, we are giving a chance to the authors to revise the article.

5. Review Comments to the Author

Reviewer #1: I must congratulate whole team for their excellent effort to put this issue in a setting with a limited resources touching upon the complex issue of demographic profiles. The issue narrates the local issue very relevantly and appears pertinent for their geographical are and definitely is issue for the policy makers so that in an era where AI is going to replace the people who do not use the AI.

The limitations in use is definitely marred by the variety of demographic and societal issues including the financial constraints. These issues have been brought by many in the literature. These issues might not be of added value to the medical students and scientist but definitely of immense help to the social engineers, policy makers and government agencies.

Also for the medical curriculum a study that explores the outcome of AI tools in the performance and enhancement of their performance will be a great help so that it can be an integral part of the curriculum.

Response to Reviewer #1: Thank you very much for your kind and encouraging feedback. We appreciate your recognition of the relevance of this study within a resource-limited setting and its potential value for policymakers and stakeholders beyond the medical field.

We also agree with your valuable insight regarding the need for future studies that evaluate the impact of AI tools on student performance. This is indeed a critical step toward informed curriculum integration, and we hope our work can serve as a foundation for such future research.

Thank you again for your thoughtful comments.

Reviewer #2: This study presents an original investigation into the awareness, attitudes, and usage of ChatGPT among Sudanese medical students using a cross-sectional survey design. The manuscript benefits from a large sample size and a clear focus on a contemporary issue in medical education. However, several concerns regarding the conceptual framing, methodology, and reporting must be addressed.

Response to Reviewer #2:

We would like to thank you for your constructive and insightful feedback. We appreciate your recognition of the originality of our study, the relevance of the topic, and the strengths in our sample size and focus.

We acknowledge the concerns raised regarding the conceptual framing, methodology, and reporting, and we have carefully reviewed each point. Below, we provide detailed responses and outline the revisions made to address your comments.

Major Comments

1. Research Gap and Objectives

The introduction does not clearly articulate the specific research gap. While the manuscript broadly discusses AI’s potential in medical education, it fails to specify what prior studies have overlooked, particularly in Sudan. Revise the introduction to explicitly define the research gap and justify the need for this study.

Response to Comment 1 – Research Gap and Objectives

Thank you for your valuable feedback. We have carefully revised the Introduction to clearly articulate the specific research gap. In the updated version, we now explicitly highlight what prior studies have addressed and what remains unexplored, particularly in the context of Sudan. We also strengthened the justification for the study by outlining the significance of exploring ChatGPT implementation within medical education in a low-resource setting.

2. Claims and Supporting Evidence

The manuscript describes ChatGPT as a transformative force in medical education and highlights its broad applicability (e.g., clinical decision support) without sufficient empirical evidence or rigorous literature support. Distinguish between potential benefits and established outcomes.

Response to Comment 2 – Claims and Supporting Evidence:

Thank you for your thoughtful feedback. We have revised the manuscript to address the concern regarding the strength of our claims. Specifically, we have modified language that previously described ChatGPT as a "transformative force" to more cautiously reflect its potential. Additionally, we have clarified the distinction between potential benefits and established outcomes, and supported our statements with more appropriate and up-to-date literature where available.

3. Methodological Design

The use of a cross-sectional design limits causal inferences. Additionally, reliance on convenience sampling may introduce selection bias, potentially over-representing students with better digital access. Acknowledge these limitations explicitly in the manuscript. Consider suggesting future research with longitudinal or randomized sampling designs. The rationale behind the use of non-parametric tests and linear regression analysis should be explicitly stated

Response:

Response to Comment 3 – Methodological Design

We thank the reviewer for this valuable comment. We have explicitly acknowledged the limitations of the cross-sectional design and convenience sampling, including their impact on causal inference and potential selection bias. The rationale for using non-parametric tests was based on the non-normal distribution of our data, while linear regression was applied to examine associations while adjusting for confounders. We also noted that future studies could benefit from longitudinal designs and randomized sampling methods.

4. Data Collection and Instrument Validation

The survey was distributed exclusively through online channels. This digital-only approach risks excluding students with poor or inconsistent internet access. Moreover, while the questionnaire was adapted from previous work, there is no mention of local reliability or validity testing (e.g., Cronbach’s alpha). Authors should provide details on any pilot testing, reliability, and validity analyses performed for the adapted questionnaire. If such analyses have not been conducted, consider including them or discussing this as a limitation.

Response to Comment 3 – Data Collection and Instrument Validation

We thank the reviewer for this feedback. We have added some lines regarding the cronbach alpha value for the subscales in the data collection tool section. Moreover, the questionnaire was adopted as it is without changes, and we have added a sentence for that.

5. Discussion and Interpretation of Findings

The discussion presents results without connecting them to prior studies.

The discussion section presents the findings largely in isolation, without a cohesive narrative that integrates methodological limitations, contextual factors (e.g., socio-economic and infrastructural challenges in Sudan), and actionable recommendations. The authors need to enhance the discussion by contextualizing your findings within the existing literature. Explore the implications of observed trends (such as gender differences) in more depth, and propose concrete strategies for addressing identified challenges in medical education and digital access.

Response to Comment 5 a– Discussion and Interpretation of Findings :

Thank you for your valuable feedback. We have revised the manuscript to address the concerns regarding the presentation of results and their integration with prior studies. Specifically, we have acknowledged the limitations of the cross-sectional design and the potential for selection bias due to convenience sampling, which is now clearly discussed in the limitations section. To enhance the robustness of our findings, we have also recommended future research using longitudinal or randomized sampling approaches to strengthen causal inferences, as stated in the final paragraph of the Discussion section.

Response to Comment 5 b– Discussion and Interpretation of Findings :

We greatly appreciate your insightful feedback. In response, we have now connected our findings to existing literature and provided more in-depth insights. However, we noted that some factors explored in our study—such as internet quality, mode of access, marital status, living status, and employment status—have not been widely addressed in previous studies, making it challenging to directly contextualize them within the existing body of research. We have discussed these gaps and the unique contributions of our study in the revised Discussion section.

Minor Comments

1.Ensure consistency in terminology when referring to ChatGPT and other AI tools.

2.Clarify any abbreviations (e.g., SDGs for Sudanese currency) when first introduced.

3.Review the overall structure to maintain a logical flow from the introduction through to the conclusions.

Response to Minor Comment 1 – Consistency in Terminology:

Thank you for pointing this out. We have reviewed the manuscript and ensured consistency in the terminology used to refer to ChatGPT and other AI tools throughout the text. All terms are now used uniformly to avoid any confusion.

Response to Minor Comment 2 – Clarification of Abbreviations:

We appreciate your feedback. We have clarified all abbreviations, including "SDGs," which refers to Sudanese currency. The full term is now defined upon first use in the manuscript to ensure clarity for readers.

Response to Minor Comment 3 – Structure and Flow:

Thank you for your suggestion. We have reviewed the overall structure of the manuscript and made adjustments to ensure a more logical flow from the introduction to the conclusions. We have made transitions smoother to improve coherence and readability throughout the paper.

The study addresses an important and timely topic in medical education. However, before further consideration, the manuscript requires revisions to clearly define its research gap, strengthen methodological reporting, and provide a more integrated discussion of the findings. Addressing these points will enhance the scientific rigor and clarity of your work.

Reviewer #3: The paper is well-written and provides valuable insights into the awareness, usage, and identification of economical and socio-demographic indicators related to the use of large language models, particularly ChatGPT. We offer several comments to enhance this work further.

1. The introduction lacks adequate citations.

Response to Reviewer #3:

Thank you for your positive comments and constructive feedback. We appreciate your recognition of the value of our study, and we are grateful for your suggestions to enhance the manuscript.

Resp

---

## [Decision Letter · Decision Letter 1]

30 Apr 2025

Shaping the Future of Medical Education: A Cross-Sectional Study on ChatGPT Attitude and Usage Among Medical Students in Sudan, 2024

PONE-D-25-05850R1

Dear Dr. Abdalmotalib,

We’re pleased to inform you that your manuscript has been judged scientifically suitable for publication and will be formally accepted for publication once it meets all outstanding technical requirements.

Kind regards,

Sujita Kumar Kar

Academic Editor

PLOS ONE

Additional Editor Comments (optional):

The article is acceptable.

Reviewers' comments:

Reviewer's Responses to Questions

**Comments to the Author**

1. If the authors have adequately addressed your comments raised in a previous round of review and you feel that this manuscript is now acceptable for publication, you may indicate that here to bypass the “Comments to the Author” section, enter your conflict of interest statement in the “Confidential to Editor” section, and submit your "Accept" recommendation.

Reviewer #1: All comments have been addressed

Reviewer #3: All comments have been addressed

2. Is the manuscript technically sound, and do the data support the conclusions?

Reviewer #1: Yes

Reviewer #3: Yes

3. Has the statistical analysis been performed appropriately and rigorously? 

Reviewer #1: Yes

Reviewer #3: Yes

4. Have the authors made all data underlying the findings in their manuscript fully available?

Reviewer #1: Yes

Reviewer #3: (No Response)

5. Is the manuscript presented in an intelligible fashion and written in standard English?

Reviewer #1: Yes

Reviewer #3: Yes

6. Review Comments to the Author

Reviewer #1: (No Response)

Reviewer #3: (No Response)

7. PLOS authors have the option to publish the peer review history of their article (what does this mean? ). If published, this will include your full peer review and any attached files.

**Do you want your identity to be public for this peer review?** For information about this choice, including consent withdrawal, please see our Privacy Policy .

Reviewer #1: No

Reviewer #3: No

---

## [Editor Report · Acceptance letter]

PONE-D-25-05850R1

PLOS ONE

Dear Dr. Abdalmotalib,

I'm pleased to inform you that your manuscript has been deemed suitable for publication in PLOS ONE. Congratulations! Your manuscript is now being handed over to our production team.

Kind regards,

on behalf of

Dr. Sujita Kumar Kar

Academic Editor

PLOS ONE